# Prevalence and associated factors of active trachoma among 1–9 years of age children in Andabet district, northwest Ethiopia, 2023: A multi-level mixed-effect analysis

Zufan Alamrie Asmare[1]*, Beminate Lemma Seifu[2], Kusse Urmale Mare[3], Yordanos Sisay Asgedom[4], Bizunesh Fantahun Kase[2], Abdu Hailu Shibeshi[5], Tsion Mulat Tebeje[6], Afework Alemu Lombebo[7], Kebede Gemeda Sabo[3], Bezawit Melak Fente[8], Achamyeleh Birhanu Teshale[9], Hiwot Altaye Asebe[2]

1 Department of Ophthalmology, School of Medicine and Health Science, Debre Tabor University, Debre Tabor, Ethiopia, 2 Department of Public Health, Collage of Medicine and Health Science, Samara University, Afar, Ethiopia, 3 Department of Nursing, College of Medicine and Health Sciences, Samara University, Afar, Ethiopia, 4 Department of Epidemiology and Biostatics, College of Health Sciences and Medicine, Wolaita Sodo University, Wolaita Sodo, Ethiopia, 5 Department of Statistics, College of Natural and Computational Science, Samara University, Afar, Ethiopia, 6 School of Public Health, College of Health Science and Medicine, Dilla University, Dilla, Ethiopia, 7 School of Medicine, College of Health Science and Medicine, Wolaita Sodo University, Wolaita Sodo, Ethiopia, 8 Department of General Midwifery, School of Midwifery, College of Medicine & Health Sciences, University of Gondar, Gondar, Ethiopia, 9 Department of Epidemiology and Biostatistics, Institute of Public Health, College of Medicine and Health Sciences, University of Gondar, Gondar, Ethiopia

* zufanalamrie2@gmail.com

## Abstract

### Background

Trachoma is the chief cause of preventable blindness worldwide and has been earmarked for elimination as a public health problem by 2030. Despite the five-year Surgery, Antibiotics, Facial cleanliness, and Environmental improvement (SAFE)-based interventions in the Andabet district, the prevalence of trachomatous follicular (TF) was 37%. With such a high prevalence of TF, the determinant factors were not revealed. Besides, there were no reports on the overall prevalence of active trachoma (i.e.TF and or trachomatous intense (TI)).

### Objective

To determine the prevalence and associated factors of active trachoma among 1–9 years of age children in the Andabet district.

### Method

A community-based cross-sectional study was conducted among children aged under nine years from March 1–30, 2023 in Andabet district, Northwest Ethiopia. Multi-stage systematic random sampling was employed to reach 540 children. A multilevel mixed-effect logistic regression analysis was employed to assess factors associated with active trachoma. We fitted both random effect and fixed effect analysis. Finally, variables with p<0.05 in the

**Data Availability Statement:** All relevant data are within the paper and its Supporting information files.

**Funding:** The author(s) received no specific funding for this work.

**Competing interests:** The authors have declared that no competing interests exist.

multivariable multilevel analysis were claimed to be significantly associated with active trachoma.

## Result

In this study, the overall prevalence of active trachoma was 35.37% (95% CI: 31.32%, 39.41%). The prevalence of TF and TI was 31.3% and 4.07% respectively. In the multilevel logistic regression analysis ocular discharge, fly-eye contact, latrine utilization, and source of water were significantly associated with the prevalence of active trachoma.

## Conclusion

In this study, the prevalence of active trachoma was much higher than the World Health Organization (WHO) threshold prevalence. Ocular discharge, fly-eye contact, latrine utilization, and source of water were independent determinants of active trachoma among children (1–9 years). Therefore, paying special attention to these high-risk groups could decrease the prevalence of a neglected hyperendemic disease, active trachoma.

### Author summary

Trachoma, caused by ocular infection with Chlamydia trachomatis, is the chief cause of preventable blindness worldwide. A key component of the WHO's elimination strategy is mass treatment with oral antibiotics, education relating to facial cleanliness, environmental improvement, and hygienic conditions. It is crucial to conduct community-based prevalence of trachoma by incorporating community-level factors to determine whether community-level interventions are necessary. A thorough understanding of the risk factors of trachoma may foster trachoma elimination efforts. The purpose of this study was to determine the prevalence and associated factors of active trachoma in andabet district by using a multi-level mixed-effect analysis. This study found that 35.37% of children (1–9 years) in the Andabet district have active trachoma in 2023, which is a public health concern. In addition, Individual and community-level factors associated with the diseases were identified. This study demonstrates the need to consider support for the introduction of those interventions for trachoma elimination in Ethiopia. This will eliminate an estimated half of the global burden.

## Introduction

Trachoma is the chief cause of preventable blindness worldwide and has been earmarked for elimination as a public health problem by 2030 [1,2]. Children aged under nine are most likely anguish from active trachoma, with prevalence rates roaming from 60–90% [3,4]. Globally, 1.3 million people shunned their sight and 1.8 million become visually impaired from the disease [5,6]. While in Ethiopia, 1.2 million people had a visual loss, and 2.8 million visual impairment cropped up by the disease [7]. It is estimated that blindness and visual impairment cost US$ 2.9–5.3 billion annually adrift from productivity, dawning to US$ 8 billion when trichiasis is embraced [8,9]. Those wrestling with visual impairment or blindness have a deteriorated quality of life [9].

The WHO notified 157.7 million people pervading in districts where active trachoma was a public health venture, 88 percent of them in Africa and half in Ethiopia (69,802,693) [2]. In Ethiopia, trachoma is the second most common cause of blindness and the third most common cause of low vision [10]. The prevalence of active trachoma among children aged 1–9 years old was 40.1% and it is ubiquitous across the country. Albeit, the Amhara region bore the highest prevalence (62.6%) [8,11,12]. Despite the perpetual endeavor in halting the quandary, it's still a public health problem in Ethiopia, particularly in the Amhara region [11].

Shreds of evidence have revealed that different sociodemographic, behavioral, and environmental factors have been associated with the prevalence of under-nine trachoma, although they differ between settings [13–16]. Of the socio-demographic factors, Age, sex, family size, and educational status, were reputed to affect the prevalence of trachoma [13,14]. Up on children's hygienic behavior, factors such as ocular discharge, nasal discharge, flies on their faces, soap used for face washing, and fomite-sharing practices have an impact on the prevalence [13,16]. Moreover, environmental factors such as the availability and utilization of latrines, waste disposal pit utilization, and scarcity of water have also been associated with the prevalence of trachoma [13,14]. Likewise, limited access to latrines boon fecal contamination of the environment, which backs fly breading, another mechanical vector for trachoma transmission [13].

Despite the five-year Surgery, Antibiotics, Facial cleanliness, and Environmental improvement (SAFE)-based interventions in the Andabet district, the prevalence of trachomatous follicular (TF) was 37% [12]. Other than the three similar settings in which SAFE was equally implemented, it remained hyperendemic and the reason was an enigma in the Andabet district. The previous study only examined TF prevalence. There were no reports on the prevalence of trachomatous intense (TI) and besides, with such a high prevalence of TF, the determinant factors were not revealed. To our knowledge, this is the first study in the district to examine the overall prevalence of active trachoma (TF and or TI) and associated factors. Moreover, most of the previous studies done out of the district did not consider the community-level factors that could affect the prevalence of the disease. But it is imperative to consider factors on an individual and community level in preventing the disease, as well as implementing policies and programs to reduce trachoma. Thus, this study aimed to determine the prevalence and associated factors of active trachoma among 1–9 years of age children using a multilevel mixed-effect analysis.

## Methods

### Ethics statement

The study has acceded to the tenets of the Declaration of Helsinki and approval was solicited and attained from the Institutional Review Board of Debre Tabor University, Health Science College (Reference Number: 2881/2023). A permission letter was procured from Andabet district administrative office. The guardians were informed that the study would not foist harm on children. There were no personal identifiers and the confidentiality of the study participants was retained at all stages of data processing. Written Informed consent was obtained from each parent/guardian. In addition, confidentiality was held by virtue of codes and dodging personal identifiers. Trachoma-infected children were referred to the closest health facility.

### Study design and settings

A community-based cross-sectional study was conducted among children aged under nine years from March 1–30, 2023. Andabet district, the study area, is sited 150 km from Bahir Dar, the capital city of Amhara National Regional State, and 717 km from Addis Ababa, the capital

of Ethiopia. The district of Andabet encompasses a large geographical area and has the preeminent population density. Based on the 2019 regional population census, the district's protruding total population is 152,683 with 34765 households verified by 26 kebeles. A primary health care center and two health posts are embedded there. There was a high prevalence of TF in the district after 8 to 11 years of implementation of SAFE.

## Study population and eligibility criteria

All children whose age was found in the range of 1–9 years in the Andabet district were the Source population and we include children from the age of 1–9 years who lived for at least 6 months in the study area. In contrariwise, children who are unable to undergo physical examination due to medical illness were excluded from the study.

## Sample size determination and sampling procedure

We appraised the required sample size using the single population proportion formula. We assumed, based on a previous similar study, an observed prevalence of active trachoma in Ebinat, Ethiopia (36.1%) [17], which we sought to estimate 95% confidence within ±5% margin of error. We used a design effect of 1.5 and allowed for a 10% non-response rate, the final sample size for this study was determined to be 585.

A multistage sampling technique was used during the sampling process. Based on a list of kebeles provided by the Andabet district administration bureau, six kebeles out of 26 kebeles were selected by using a simple random sampling method. To determine the required sample size for each randomly selected kebele, population proportional allocation was employed.

In the selected kebele, there were only 4785 households with at least one child between the ages of 1 and 9. Systematic random sampling with an interval of 8 was used to select households with children between the ages of 1 and 9. Before starting the sampling, pen spinning was carried out to mark the starting point of the village. In the case of more than one child aged 1–9 years per household, one child was selected using the lottery method.

## Operational definition

**Active trachoma.**   The presence of Trachomatous inflammation, follicles/TF (the appearance of five or more follicles with a diameter of greater than 0.5 mm in the central part of the upper tarsal conjunctiva), and/or Trachomatous inflammation intense/TI (pronounced inflammatory thickening of the tarsal conjunctiva that obscures more than half of the normal deep tarsal vessels) on one or both eyes [18].

**Community level of women illiteracy.**   It is the clumped community-level variable derived from maternal educational level and rated as the proportion of women with no formal education at the kebele/community level. Based on a median value it was then divided into low (mothers from communities with lower illiteracy levels) and high (mothers from communities with higher illiteracy levels) categories [19,20].

**Latrine utilization.**   Those latrines with at least two of these: the presence of a splash of urine, fresh excreta inside the latrine, footpath to the latrine, and the absence of a spider web of the squat were considered utilized [21].

**Waste disposal pit utilization.**   Those pits with at least one of these: domestic products, discarded unwanted agricultural products, or ashes (a burned sign of waste) were considered utilized [22].

## Data collection tools and procedures

After reviewing the available literatures, the data collection tool was developed. A pretested structured questionnaire, observational checklists, and a physical examination were used to collect data. There were four parts to the questionnaire: sociodemographic variables, child behavioral variables, environmental-related variables, and observation checklists. The environmental and household data were collected by three trained ophthalmic nurses. Through the use of 2.5x loupes, two Trachomatous trichiasis (TT)-trained surgeon nurses certified for trachoma grading assessed each child for active trachoma signs. In accordance with the WHO simplified grading system, certified TT-trained trachoma grader surgeons examined both eyes for active trachoma. Using cotton tip applicators and alcohol for hand disinfection, an aseptic eyelid eversion was performed on the children.

## Data quality control

To ensure consistency, the data collection tool was first developed in English, then translated into the local language (Amharic), and then back to English. Then, to accustom data collectors and supervisors to the data collection procedures, two days of training were provided. Lastly, a pre-test was conducted on 5% of the total sample size in another kebele that was not included in the study. Unclear questions were edited and modified based on the analysis of the pre-test. Data were evaluated by supervisors and investigators for completeness, accuracy, and clarity.

## Data processing and analysis

Epi-Data version 4.6 was used for data entry, followed by STATA 16 for cleaning, coding, and analyzing the data. Text, tables, and figures were used to report the descriptive statistics. The prevalence of active trachoma with its 95% Confidence interval (CI) was reported. A multilevel logistic regression analysis was employed to assess factors associated with active trachoma to account for the hierarchical nature of the data in which children were nested within-cluster and children within the same cluster are more likely to share similar characteristics than children in another cluster which contravenes the independent assumptions of the standard logistic regression model such as the independent and equal variance assumptions.

First bivariable multilevel logistic regression analysis was executed and those variables with p-value <0.20 were considered for multivariable multi-level analysis. While performing a multilevel binary logistic regression analysis, we fitted both random effect and fixed effect analyses. The random effect parameter, intraclass correlation coefficient (ICC) computes the degree of heterogeneity in the prevalence of active trachoma between clusters and an ICC of more than 10% indicates that accounting for the cluster-level variability of active trachoma using multilevel analysis is relevant. In addition, proportion change in variance (PCV), and median odds ratio (MOR) were appraised. Moreover, multicollinearity was verified using the variance inflation factor (VIF) and we obtained a VIF of less than five for each independent variable with a mean VIF of 1.90, denoting there was no significant multicollinearity between independent variables.

In fixed effect analysis, four models were fitted; null model (without explanatory variables), model 1 (containing only individual-level factors), model 2 (examining the effect of community-level factors), and model 3 (which incorporates both individual and community-level factors simultaneously). Among the four models fitted, the last model (model 4) was selected as the best-fitted model given that it has the lowest deviance and highest PCV. The adjusted odds ratio (AOR) with its 95% CI was reported for all models fitted. Interpretations, however, are based on the final model, the best-fit model. Finally, variables with p<0.05 in the multivariable multilevel analysis were claimed to be significantly associated with active trachoma.

## Result

### Socio-demographic characteristics of study participants

A total of 540 children in the age range of 1–9 years were included in the study providing a response rate of 92.3%. Of these, more than two-thirds (75.74%) were between the age of 4 and 9 with an overall mean age of 6.5±1 years. Almost half (50.93%) of the children were males and 252 (46.67%) were from rural areas. Regarding the household size, more than half (62.22%) of children were from a household size of 4 or less (Table 1).

### Environmental characteristics

The majority (85.16%) of the households had access to a covered pit latrine. Of them, 433 (80.19%) utilize the latrine. Almost all (97.07%) hadn't handwashing facilities near to latrine and 255(47.22%) of them didn't have separate places for animal dwellings (Table 2).

### Childhood behavioral characteristics

Of 540 children, more than half (61.67%) had fly-eye contact and 238(44.07%) had nasal discharge. The majority (92.04%) of children didn't use soap for face washing and around two-thirds (72.78%) of them didn't utilize fomite (Table 3).

**Table 1. Socio-demographic characteristics of study participants in Andabet, northwest Ethiopia, 2023: A multi-level mixed-effect analysis (n = 540).**

| Variables | Category | Frequency | Percentage |
|---|---|---|---|
| Age of the child | < 4 years | 131 | 24.26 |
| | 4 to 9 years | 409 | 75.74 |
| Sex of the child | Male | 275 | 50.93 |
| | Female | 265 | 49.07 |
| Residence | Rural | 252 | 46.67 |
| | Urban | 288 | 53.33 |
| Religion | Orthodox | 493 | 91.30 |
| | Muslim | 47 | 8.70 |
| Age of the mother | 15–24 | 76 | 14.07 |
| | 25–34 | 273 | 50.56 |
| | 35 yrs. and above | 191 | 35.37 |
| Educational level of the father | No formal education | 265 | 49.07 |
| | Primary | 140 | 25.93 |
| | Secondary &above | 135 | 25 |
| Occupation of mother | Farmer | 238 | 44.07 |
| | Housewife | 160 | 29.63 |
| | Government employee | 32 | 5.93 |
| | Merchant | 74 | 13.70 |
| | Daily laborer | 36 | 6.67 |
| Educational level of the mother | No formal education | 328 | 60.74 |
| | Primary | 127 | 23.52 |
| | Secondary &above | 85 | 15.74 |
| Family size | < = 4 | 336 | 62.22 |
| | > 4 | 204 | 37.78 |
| Community-level of women's illiteracy | High | 185 | 34.26 |
| | Low | 355 | 65.74 |

**Table 2. Environmental characteristics of the households in Andabet, northwest Ethiopia, 2023: A multi-level mixed-effect analysis (n = 540).**

| Variables | Category | Frequency | Percentage |
|---|---|---|---|
| Source of water | River | 274 | 50.74 |
| | Household tap | 266 | 49.26 |
| Utilization of latrines | Yes | 433 | 80.19 |
| | No | 107 | 19.81 |
| Utilization of waste disposal pit | Yes | 240 | 44.44 |
| | No | 300 | 55.56 |
| Type of latrine used | Covered pit | 373 | 85.16 |
| | Uncovered pit | 65 | 14.84 |
| Disposing infant faces to the latrine | Yes | 431 | 79.81 |
| | No | 109 | 20.19 |
| Availability of handwashing material near to latrine | Yes | 13 | 2.93 |
| | No | 431 | 97.07 |
| Availability of separate animal dwelling | Yes | 285 | 52.78 |
| | No | 255 | 47.22 |

## Prevalence of active trachoma

Of all 540 children examined for the presence or absence of trachoma in their eyes, 191 children were positive for active trachoma. The overall prevalence of active trachoma was 35.37% (95% CI: 31.32%, 39.41%). The prevalence of TF and TI was 518(31.3%) and 22(4.07%) respectively (Fig 1).

## Random effect and model fitness

Table 4 revealed that in the null model, about 35.7% of the total variation in the prevalence of active trachoma occurred at the cluster (kebele) level and is attributable to the community-level factors. In addition, the null model also had the highest MOR value (3.59) indicating when randomly selecting children from one kebele with a higher risk of active trachoma and the other kebele at lower risk, children at the kebele with a higher risk of active trachoma had

**Table 3. Childhood behavioral characteristics in Andabet, northwest Ethiopia, 2023: A multi-level mixed-effect analysis (n = 540).**

| Variables | Category | Frequency | Percentage |
|---|---|---|---|
| Nasal discharge | Yes | 238 | 44.07 |
| | No | 302 | 55.93 |
| Ocular discharge | Yes | 145 | 26.85 |
| | No | 395 | 73.15 |
| Fly-eye contact | Yes | 333 | 61.67 |
| | No | 207 | 38.33 |
| Utilization of fomites | Yes | 147 | 27.22 |
| | No | 393 | 72.78 |
| Utilization of soap | Yes | 43 | 7.96 |
| | No | 497 | 92.04 |
| Frequency of face washing | Once daily | 22 | 4.07 |
| | Twice and above | 518 | 95.93 |
| Hand washing before the face | Yes | 532 | 98.52 |
| | No | 8 | 1.48 |

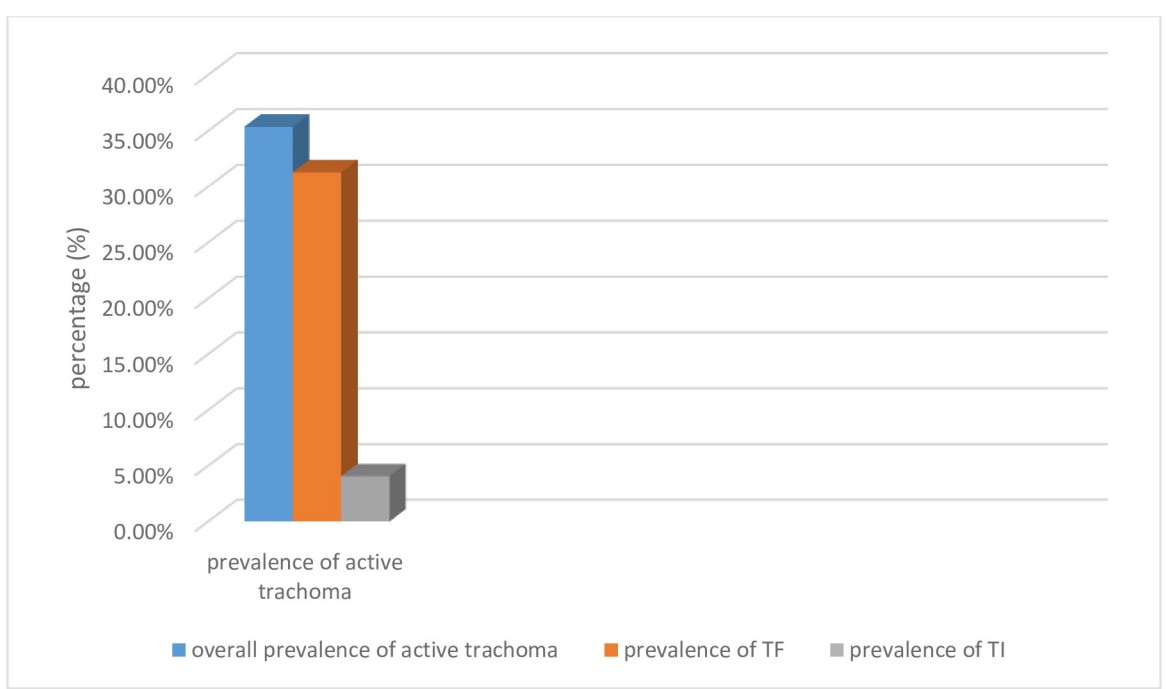

**Fig 1. Prevalence of active trachoma among 1–9 years of age children in Andabet district, northwest Ethiopia, 2023: A multi-level mixed-effect analysis.**

3.59 times higher odds of having active trachoma as compared with their counterparts. Furthermore, the highest (85%) PCV in the full model (model 3), indicates that 85% of the community-level variation in the prevalence of active trachoma was explained by the combined factors at both the individual and community levels. The model fitness was done using deviance in which the final model (model 3) was the best-fitted model since it had the lowest deviance.

## Factors associated with active trachoma

In multivariable multilevel logistic regression analysis, where both the individual and community level factors were fitted simultaneously; ocular discharge, fly-eye contact, latrine utilization, and source of water were significantly associated with the prevalence of active trachoma.

Children who had ocular discharge had 2.01 (AOR = 2.01; 95%CI: 1.11, 3.64) times higher odds of developing active trachoma as compared to those children who had no ocular discharge. Regarding fly-eye contact, children who had fly-eye contact were 1.96 (AOR = 1.96;

**Table 4. Random effect and model fitness in prevalence and associated factors of active trachoma among 1–9 years of age children in Andabet district, northwest Ethiopia, 2023: A multi-level mixed-effect analysis (n = 540).**

| Parameter | Null model | Model 1 | Model 2 | Model 3 |
|---|---|---|---|---|
| Log likelihood | -247.5 | -246.2 | -272.8 | -244.48 |
| MOR | 3.59 | 2.63 | 3.28 | 1.61 |
| PCV | Ref. | 0.43 | 0.12 | 0.85 |
| ICC | 0.357 | 0.239 | 0.326 | 0.074 |
| deviance | 495.0 | 492.4 | 545.6 | 488.8 |

**Table 5. Multi-level mixed-effect logistic regression analysis for factors associated with active trachoma among 1–9 years of age children in Andabet district, Northwest Ethiopia, 2023 (n = 540).**

| Variables | Null model | Model 1 AOR 95%(CI) | Model 2 AOR 95%(CI) | Model 3 AOR 95%(CI) |
|---|---|---|---|---|
| Age of the child | | | | |
| < 4 years | | 0.80 (0.46,1.38) | | 0.81 (0.47,1.39) |
| 4 to 9 years | | 1.00 | | 1.00 |
| Nasal discharge | | | | |
| Yes | | 0.85 (0.46,1.56) | | 0.89 (0.50,1.61) |
| No | | 1.00 | | 1.00 |
| Ocular discharge | | | | |
| Yes | | 2.15 (1.2,3.8)* | | 2.01 (1.11,3.64) * |
| No | | 1.00 | | 1.00 |
| Fly-eye contact | | | | |
| Yes | | 1.89 (1.06,3.38)* | | 1.96 (1.09,3.53) * |
| No | | 1.00 | | 1.00 |
| Family size | | | | |
| < = 4 | | 1.00 | | 1.00 |
| >4 | | 1.56 (0.97,2.50) | | 1.59 (0.98,2.58) |
| latrine utilization | | | | |
| Yes | | 1.00 | | 1.00 |
| No | | 5.39 (2.91,9.97) ** | | 5.28 (2.88,9.70)** |
| Utilization of waste disposal pit | | | | |
| Yes | | 1.00 | | 1.00 |
| No | | 1.55 (0.88,2.73) | | 1.66 (0.95,2.89) |
| Residence | | | | |
| Rural | | | 0.95 (0.37,2.41) | 0.87 (0.37,2.07) |
| Urban | | | 1.00 | 1.00 |
| Source of water | | | | |
| River | | | 2.67 (0.52,13.60) | 3.89 (1.30,11.67)* |
| Household tap | | | 1.00 | 1.00 |
| Community-level of women's illiteracy | | | | |
| High | | | 0.46 (0.10,2.09) | 1.07 (0.37,3.03) |
| low | | | 1.00 | 1.00 |

Note:

** P<0.01, and

* = P<0.05

95%CI: 1.09, 3.53) times higher odds of developing active trachoma as compared to those children who had no fly-eye contact. In the same manner, the odds of developing active trachoma among children aged 1–9 years from families who didn't utilize latrine were 5.28 (AOR = 5.28; 95%CI: 2.88,9.70) times higher than children from families who utilize latrine. On the source of water, the odds of developing active trachoma among children aged 1–9 years from households who get water from the river had 3.89 (AOR = 3.89; 95%CI: 1.30,11.67) times higher than children from households who get water from the household tap (Table 5).

## Discussion

The study sought to assess the prevalence and associated factors of active trachoma in the Andabet district, Northwest Ethiopia. This study revealed the prevalence of active trachoma

among 1–9 years was 35.37%. This finding is consistent with different studies done in Areka, Zala district, Ethiopia, Nigeria, Chad, Uganda, Central African Republic, and Senegal [23–28]. However, this prevalence of active trachoma was found lower compared to different studies conducted in Ankober, Amhara, Amaro, Burji, and Horo Guduru [16,29–31] and higher than different studies conducted in different countries [11,13,14,32]. The discrepancy might be due to the difference in the ground status of trachoma prevention practice, study setting, period, and intervention. Besides, the availability and accessibility of health facilities, as well as the capacity of water, sanitation, and hygiene, differs between countries [33,34]. Moreover, the discrepancy of this finding with that of the findings of studies conducted out of Ethiopia might be due to socio-demographic and cultural differences.

In this study, we found that children who had ocular discharge were more likely to develop active trachoma as compared to those children who had no ocular discharge. This finding is supported by studies done in southern and northern Wollo zone districts, Dangila, Gambia, and Tanzania [13,35–37]. which similarly showed children with ocular discharge were more likely for active trachoma infection. This might be because a discharge from the infected eye prompts transmission of infection by direct contact or via fingers, flies, or fomites [38].

Fly-eye contact was another factor for the prevalence of active trachoma among [1–9] in which children who had fly-eye contact were more likely to develop active trachoma than those children who had no fly-eye contact. This finding was similar to those of studies done in Dangila, Rural Ethiopia, and Ankober [13,29,37], which similarly showed fly-eye contact as a risk factor for trachoma. This might be because the eye-questing flies Musca sorbents and other domestic Muscidae are vectors for Chlamydia trachomatis and open the trachoma transmission route [39].

Consistent with other studies conducted in Baso-Liben, Ankober districts, and Dangila [13,29,32] in this study, children (1–9 years) who lived in households that didn't utilize latrines were more likely to develop active trachoma than children (1–9 years) who lived in households that utilize latrine. The possible reason might be due to Musca sorbents; these are a reservoir of the causative agent, Chlamydia trachomatis, that have been shown to preferentially breed in human excreta [13]. Hence, employing open defecation beside the house is a favorable environment for breeding Musca sorbents and an indispensable benefactor to disease transmission.

The fourth important finding in this study is the source of water which is a community-level factor associated with the prevalence of active trachoma among children (1–9 years). That is, Children (1–9 years) living in households who get water from the river were more likely to develop active trachoma than children from households who get water from the household tap. This finding is supported by different studies done in Waghemera and Madda Walabu [40,41]. This might be due to a river, or any other unprotected source of water that can serve as a reservoir of infection because it is a breeding ground for flies, as well as a habitat for Chlamydia trachomatous [42].

## Strengths and limitations of the study

There were strengths and limitations in this study. To begin with the strength, this study investigated neglected tropical diseases in children aged 1–9 years under WHO guidelines. Besides, the study uses multilevel modeling that takes into account the clustering effect to draw valid conclusions and inferences. Moreover, a sufficient sample size was used to ascertain representativeness. This study, however, has limitations due to its cross-sectional nature. It may not show a true temporal relationship between the outcome and the independent variables.

Besides, there might be an anticipation of social desirability bias and potential recall bias while assessing ticklish variables.

## Conclusion

In this study, the prevalence of active trachoma was much higher than the WHO threshold prevalence. It's still a severe public health problem and far from the elimination of trachoma as a public health problem in this community. Ocular discharge, fly-eye contact, latrine utilization, and source of water were independent determinants of active trachoma among children (1–9 years).

Therefore, an intervention area needs to be refined for personal hygiene-related activities such as washing children's faces utterly to remove dirt (ocular discharge), and fly-eye contact. Significant emphasis and framework are crucial to the construction and service provision of household taps. Besides, the building and use of latrines need to be prioritized.

## Supporting information

**S1 Data. All relevant data are within the paper and its Supporting Information files.** (XLSX)

## Acknowledgments

We are grateful to the Debre Tabor University, study participants, and data collectors.

## Author Contributions

**Conceptualization:** Zufan Alamrie Asmare.

**Data curation:** Zufan Alamrie Asmare.

**Formal analysis:** Zufan Alamrie Asmare.

**Investigation:** Zufan Alamrie Asmare.

**Methodology:** Zufan Alamrie Asmare.

**Project administration:** Zufan Alamrie Asmare.

**Resources:** Zufan Alamrie Asmare.

**Software:** Zufan Alamrie Asmare, Beminate Lemma Seifu, Achamyeleh Birhanu Teshale.

**Supervision:** Zufan Alamrie Asmare.

**Validation:** Zufan Alamrie Asmare, Beminate Lemma Seifu, Kusse Urmale Mare, Yordanos Sisay Asgedom, Bizunesh Fantahun Kase, Abdu Hailu Shibeshi, Tsion Mulat Tebeje, Afework Alemu Lombebo, Kebede Gemeda Sabo, Bezawit Melak Fente, Achamyeleh Birhanu Teshale, Hiwot Altaye Asebe.

**Visualization:** Zufan Alamrie Asmare, Beminate Lemma Seifu, Kusse Urmale Mare, Yordanos Sisay Asgedom, Bizunesh Fantahun Kase, Abdu Hailu Shibeshi, Tsion Mulat Tebeje, Afework Alemu Lombebo, Kebede Gemeda Sabo, Bezawit Melak Fente, Achamyeleh Birhanu Teshale, Hiwot Altaye Asebe.

**Writing – original draft:** Zufan Alamrie Asmare.

**Writing – review & editing:** Zufan Alamrie Asmare.

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
