## [Decision Letter · Decision Letter 0]

16 Jul 2023

Dear Miss Asmare,

Thank you very much for submitting your manuscript "Prevalence and associated factors of active trachoma among 1-9 years of age children in Andabet district, northwest Ethiopia, 2023: a multi-level mixed-effect analysis" for consideration at PLOS Neglected Tropical Diseases. As with all papers reviewed by the journal, your manuscript was reviewed by members of the editorial board and by several independent reviewers. The reviewers appreciated the attention to an important topic. Based on the reviews, we are likely to accept this manuscript for publication, providing that you modify the manuscript according to the review recommendations. 

Sincerely,

Joseph M. Vinetz

Section Editor

Joseph Vinetz

Section Editor

Reviewer's Responses to Questions

**Key Review Criteria Required for Acceptance?**

**Methods**

-Are the objectives of the study clearly articulated with a clear testable hypothesis stated?

-Is the study design appropriate to address the stated objectives?

-Is the population clearly described and appropriate for the hypothesis being tested?

-Is the sample size sufficient to ensure adequate power to address the hypothesis being tested?

-Were correct statistical analysis used to support conclusions?

-Are there concerns about ethical or regulatory requirements being met?

Reviewer #1: (No Response)

**Results**

-Does the analysis presented match the analysis plan?

-Are the results clearly and completely presented?

-Are the figures (Tables, Images) of sufficient quality for clarity?

Reviewer #1: (No Response)

**Conclusions**

-Are the conclusions supported by the data presented?

-Are the limitations of analysis clearly described?

-Do the authors discuss how these data can be helpful to advance our understanding of the topic under study?

-Is public health relevance addressed?

Reviewer #1: (No Response)

**Editorial and Data Presentation Modifications?**

Reviewer #1: (No Response)

**Summary and General Comments**

Reviewer #1: This is a very improved version of the manuscript that needs only very minor grammatical corrections as listed below:

In the Abstract:

Background:

Fifth line - "was 37%. Then space and capital "W" for With.

Seventh line - TF mentioned in full above in fourth and fifth lines but TI not mentioned in full. Plus ass TI in full with Acronym in seventh line as some people may read only the abstract.

Conclusion:

First line - Write WHO in full then the Acronym as some people may read only abstract.

Fourth line - use "paying" instead of "taking", reads better as Therefore, paying special attention .......".

In the Main body:

Abstract: I suggest same corrections as above in Lines 38, 40, 57 and 59.

Line 65: synchronise with Line 57; write in full with Acronym in bracket in line 57 and then write WHO's in line 65.

Line 144: I am not familiar with the term "pen spiring". Please cross check and amend if necessary.

Line 293: Better to use "because " instead of "due to" so the sentence reads well; "This might be because a discharge from the infected eye prompts transmission of infection by direct contact or via fingers, flies, or fomites (38)".

Line 299: Better to use "because " instead of "due to" so the sentence reads well.

Line 300: Use "are" instead of "is" reference is to "eye-questing flies and other domestic Muscidae. Also, use "for" instead of "in" and plural "open" instead of singular "opens as I believe reference here is to the word "vectors". Sentence reads better (Lines 299 and 300) as: This might be because the eye-questing flies Musca sorbents and other domestic Muscidae are vectors for Chlamydia trachomatis and open the trachoma transmission route (39).

Line 305: Please add "that have" to "been" so the sentence (Lines 304 and 305) reads well; The possible reason might be due to Musca sorbents; these are a reservoir of the causative agent, Chlamydia trachomatis, that have been shown to preferentially breed in human excreta (13).

Line 316: Remove the full stop before "There".

PLOS authors have the option to publish the peer review history of their article (what does this mean?). If published, this will include your full peer review and any attached files.

Reviewer #1: No

Figure Files:

Data Requirements:

Reproducibility:

References

---

## [Editor Report · Decision Letter 1]

5 Aug 2023

Dear Miss Asmare,

We are pleased to inform you that your manuscript 'Prevalence and associated factors of active trachoma among 1-9 years of age children in Andabet district, northwest Ethiopia, 2023: a multi-level mixed-effect analysis' has been provisionally accepted for publication in PLOS Neglected Tropical Diseases.

Best regards,

Joseph M. Vinetz

Section Editor

Joseph Vinetz

Section Editor

---

## [Editor Report · Acceptance letter]

14 Aug 2023

Dear Miss Asmare,

We are delighted to inform you that your manuscript, "Prevalence and associated factors of active trachoma among 1-9 years of age children in Andabet district, northwest Ethiopia, 2023: a multi-level mixed-effect analysis," has been formally accepted for publication in PLOS Neglected Tropical Diseases.

Best regards,

Shaden Kamhawi

co-Editor-in-Chief

Paul Brindley

co-Editor-in-Chief
